# The Platelet Concentrates Therapy: From the Biased Past to the Anticipated Future

**DOI:** 10.3390/bioengineering7030082

**Published:** 2020-07-30

**Authors:** Tomoyuki Kawase, Suliman Mubarak, Carlos Fernando Mourão

**Affiliations:** 1Division of Oral Bioengineering, Institute of Medicine and Dentistry, Niigata University, Niigata 950-0000, Japan; 2Department of Prosthodontics, Faculty of Dentistry, International Sudan University, Khartoum 79371, Sudan; mubarakoo86@hotmail.com; 3Department of Oral Surgery, Dentistry School, Fluminense Federal University, Rio de Janeiro 05407-002, Brazil; mouraocf@gmail.com

**Keywords:** platelet-rich plasma (PRP), platelet-rich fibrin (PRF), platelet concentrates, bone regeneration, randomized controlled trial, standardization, quality assurance

## Abstract

The ultimate goal of research on platelet concentrates (PCs) is to develop a more predictable PC therapy. Because platelet-rich plasma (PRP), a representative PC, was identified as a possible therapeutic agent for bone augmentation in the field of oral surgery, PRP and its derivative, platelet-rich fibrin (PRF), have been increasingly applied in a regenerative medicine. However, a rise in the rate of recurrence (e.g., in tendon and ligament injuries) and adverse (or nonsignificant) clinical outcomes associated with PC therapy have raised fundamental questions regarding the validity of the therapy. Thus, rigorous evidence obtained from large, high-quality randomized controlled trials must be presented to the concerned regulatory authorities of individual countries or regions. For the approval of the regulatory authorities, clinicians and research investigators should understand the real nature of PCs and PC therapy (i.e., adjuvant therapy), standardize protocols of preparation (e.g., choice of centrifuges and tubes) and clinical application (e.g., evaluation of recipient conditions), design bias-minimized randomized clinical trials, and recognize superfluous brand competitions that delay sound progress. In this review, we retrospect the recent past of PC research, reconfirm our ultimate goals, and discuss what will need to be done in future.

## 1. Introduction

Since the first study on bone regeneration [1], the aim of platelet concentrate (PC) research has been to establish a basis for the safer, more effective, and more predictable clinical use of PC therapy. Nevertheless, fundamental questions regarding the clinical efficacy of PCs have been raised increasingly over the past two decades, to the point that PC therapy has been called disappointing and a “mirage”, “miracle”, or “myth” [2]. This may be due to the rapid global spread of PCs without having first established robust evidence. To our knowledge, their clinical use has not yet been endorsed convincingly by either observational or interventional studies.

Nowadays, PC therapies have been widely applied in the fields of oral surgery, orthopedic surgery, plastic surgery, and dermatology, such as treatments for alveolar bone defects, acute injuries of muscles, tendon and ligament injuries, joint injuries, osteoarthritis, skin rejuvenation, scars, inflammation reduction, hair loss, postsurgical repair, and others. If we continue to accept PC therapy “consciously or unconsciously” in its present form, it may soon fall out of use or be remembered in medical history as “a meaningless alternative medicine”. Therefore, we must first obtain clear evidence of the functionality of PC therapy. In this review, we look back on the recent history of PC research, reconfirm the ultimate goals of this field, and discuss a plan of action for the future.

## 2. Unstable Positioning of PC Therapy in Regenerative Medicine

With its widespread application across several fields, the reliability and clinical use expectations of PC therapy seem to be declining. One reason for a decline in its performance could be due to its irregular clinical induction process. Factory-made medicines, including biologicals, are generally subjected to rigorous examination in preclinical studies, such as in vitro and in vivo animal experiments, before being tested in clinical studies to determine their safety to efficacy [3]. In contrast, in the case of PCs, since platelet-rich plasma (PRP) was already in use as a glue in surgical operation at the time of PC therapy development [4], PRP was directly tested in humans without conducting a step-by-step preclinical study to determine its feasibility and validity [5]. In addition, PRP is generally a “home-made” product: it is prepared from autologous blood samples upon request and is immediately used in clinical settings. Thus, unlike other “factory-made” (industrial) medicinal products, preclinical testing was not initially required for PCs to receive regulatory approval. This has resulted in the widespread application of PC therapy without sufficient evidence from preclinical studies to corroborate its effectiveness.

The lack of a preclinical background has led clinicians to misunderstand the applications of PC therapy. However, some clinicians do recognize PC therapy as alternative medicine, also called “complementary medicine” in Europe and categorized into “traditional medicine” by the World Health Organization (WHO) [6]. Based on this definition, PC therapy can be distinguished from conventional alternative medicine. In this way, PCs can be used as an alternative for expensive factory-made medicines, especially in low- and middle-income countries. In fact, clinicians in these countries expect favorable clinical outcomes from PC therapy and are eager to obtain any relevant information due to its commercial availability and cost-benefit ratio.

The main reasons for PC therapy not being established as an approved regenerative therapy include: (1) an unidentified and complex mechanism of action; (2) large individual variations (cases and PC quality); and, (3) poor or irreproducible clinical outcomes upon solo application. Thus, PC therapy is recognized as an “adjuvant therapy” [7,8,9]. Even though PCs have weak effects on tissue regeneration alone, these biomaterials may be able to maximize the effectiveness of primary or initial therapy, such as surgical operation or medication, as observed in the potentiation of the immune responses to antigens [10]. However, if PC therapy is to be considered a primary treatment, these surgical interventions will have to be interpreted as “conditioning therapy” to eliminate factors that interfere with the action of PCs. Alternatively, PC therapy could be considered “replacement therapy”. In either case, PCs provide elements that are required for tissue regeneration, including growth factors and scaffolding materials, which cannot be directly provided by surgical operation or medication.

We express the relationship between PC therapy and other interventions by the formulae shown below. In the case of conventional surgical operation or medicine, regardless of the systemic or local conditions, (a) these therapies (x) should work alone or in cooperation with spontaneous healing activity. Thus, this relationship could be expressed as “addition”, as shown in formula (1). In contrast, because PC action (x) is more or less influenced directly by coupling therapy or spontaneous regenerative activity (a), this relationship could be expressed as “multiplication”, as shown in formula (2). The resulting clinical outcomes are expressed as “y”.
surgical operation or medicine: y = **a** + x(1)
PC: y = **a** × x(2)

In addition, a new concept of “coupling therapy” has recently been proposed, which is based on a tissue engineering triangle [11,12] and suggests the need for stem cell replacement [13,14,15,16].

## 3. History of PC Study

In the past two decades, many efforts have been made in order to support PC therapy. However, how much these studies positively and constructively contributed to improving our understanding of PCs or the development of PC therapy is questionable. In this section, we summarize the history of PC studies, particularly with regards to its initial investigation, into four generations (Table 1).

The first generation [4] started from the first clinical report in dentistry by Marx [1], where the primary purpose was to confirm the concentrated platelets and their growth factors in PRP preparations, as indicated by the theoretical evaluation [1,17,18,19]. Simultaneously, the double-spin preparation protocol for small scale samples was optimized [18,20]. For large scale samples, several automated PRP preparation devices (machines and kits) were developed, launched, and used in clinical settings [21,22,23]. Thus, this generation of research successfully established the fundamental elements of PC study and therapy. An essential factor of first generation PC research is the addition of anticoagulants in blood by-products [e.g., acid-citrate-dextrose-A (ACD-A)] [20,24].

In the second generation studies, several PRP derivatives were developed by modifying the conventional protocols. The most successful example would be platelet-rich fibrin (PRF) developed by Choukroun [25]. In the first generation (and even at present), PRP was prepared from whole-blood samples in the presence of anticoagulants and clotted by the addition of coagulation factors, if necessary. These complicated processes require skilled operators, as the addition of anticoagulants and/or coagulation factors can have harmful effects on tissue regeneration. PRF can be prepared simply by stimulating the intrinsic coagulation pathway without the aid of anticoagulants or coagulation factors. This preparation protocol also enabled clinicians to save time and labor for PC preparation. At this point, the development of PRF was epoch-making in the history of PC research. The development of freeze-dried PRP can be included in this generation. Freeze-dried platelets were originally developed for military purposes [4]. Kawase and co-workers applied this concept and technology in PRP preparation and proposed a novel type of PRP derivative to realize a stable supply of high-efficacy PRP prepared from allogeneic blood samples [26,27]. However, mainly due to economical and safety concerns, this proposal was never considered or accepted for further pre-clinical or clinical study. Recently, a novel technology enabled the generation of platelets from induced pluripotent stem cells (iPSC) [28,29]. Thus, when PRP can be reconstituted by combining iPSC-derived platelets with major plasma components, including coagulation factors, the applicability of allogeneic PRP will be reconsidered more seriously. iPSC-derived PRP is expected to be a “game changer” in the development of PC therapy and provide more predictable therapy in combination with allogeneic stem cells, as described below.

After that, mainstream PC research became more diverse, and progressed independently. In the third generation, much attention was paid to the comparisons between PRP derivatives. In particular, the ability of PRP to retain and release growth factors and the distribution of platelets and leukocytes have been vigorously investigated [30,31,32,33,34]. It is essential and necessary to compare individual PCs from a neutral standpoint in order to optimize and standardize the preparation protocol. However, fair, comprehensive comparisons among at least four major derivatives (L-PRF, A-PRF, CGF, L-PRP, PRGF, etc.) have rarely been performed. Instead, comparisons have usually been performed among representative brands from a commercial standpoint. We hope that this trend will soon be amended to allow for advances in PC research.

The fourth generation comprises research on the tissue engineering triangle. Most conventional PC derivatives do not contain a sufficient number of circulating mesenchymal stem cells or CD34^+^ hemopoietic stem cells, if any, to induce their regenerative activities. However, because PC derivatives are able to provide both growth factors and scaffolding materials, such as easily-degradable fibrin, the triangle can be rebuilt by adding the appropriate stem cells. According to this concept, bone marrow- and adipose-derived stem cells have been examined in several preclinical studies [16,35,36,37]. However, at the levels of clinical research and clinical practice, such combinational treatments are limited [38,39]. The most successful example is the combination of tissue-engineered periosteal sheets and PRP. Strictly speaking, periosteal sheets contain some pluripotent stem cells (<0.5%), where the majority of periosteal cells are immature osteoblast progenitor cells. Therefore, although this combination was composed of progenitor cells, but not stem cells, and PRP, it resulted in remarkable outcomes to the recovery of well-balanced bone metabolism and new bone formation [40,41].

Therefore, in the past two decades, the clinical and pre-clinical PC studies have always been conducted prior to basic PC studies and have examined the effectiveness of PC therapy in a wide variety of clinical cases. However, a milestone in the history of PCs has yet to be reached and it will be difficult to attain without supporting evidence obtained from basic studies.

### 3.1. Direction and Goal of Initial Generations

Each generation was analyzed in detail. According to basic procedures, factory-made medicines are initially examined and screened in preclinical studies prior to clinical studies. In the case of PCs, clinical study has always preceded preclinical studies, such that a fundamental study is now needed to provide evidence to support its clinical use. In the first generation, a fundamental study successfully examined concentrated platelets and growth factors in PRP preparations. To further optimize the preparation protocol, especially when using the double-spin method [18,20], many studies were conducted while using various centrifugal conditions (speed and time).

However, a sharp buffy coat was formed for higher platelet recovery due to a poor understanding of PC blood fractionation and platelet behavior. At present, slow spin appears to be more efficient in the recovery of platelets [42]. Under fast spin conditions, platelets are fractionated at higher densities, along with leukocytes, and activated to form platelet-platelet aggregates and platelet-leukocyte aggregates. In addition, a significant number of platelets are incorporated into red blood cell fractions. As a result, the platelet concentration of PRP after the second spin is unable to reach the high levels expected. In terms of growth factors, platelets activated in the process of centrifugation release growth factors, resulting in decreased growth factor levels. Therefore, although relatively faster centrifugation speeds were recommended in this era, this led to platelet aggregation, growth factor loss, and concentrated leukocyte inclusion [43,44,45,46].

Another point of consideration is leukocyte inclusion [47,48,49]. This was not initially discussed; however, Marx’s opinion that the ideal concentration rate of platelets was 3–4-fold in PRP preparations suggests a need to avoid not only highly concentrated platelets, but also leukocyte inclusion [23,24]. To date, this has been a topic of debate: some researchers claim that leukocytes should be included in order to facilitate wound debridement, wound healing, and subsequent tissue regeneration [47,50], whereas some are concerned about the unexpected exacerbation of inflammation [51]. In regenerative cartilage therapy, the exclusion of leukocytes seems to yield better outcomes [52,53]. For other applications, further investigation will be needed in order to reach conclusions.

### 3.2. Development of Automated PRP Preparation Devices

In the first generation, automated PRP preparation devices and kits were developed [23]. Due to their structural limitations, automated machines (e.g., GPSIII platelet concentration system) (Zimmer Biomet, Warsaw, IN, USA) (Figure 1a) were developed for the preparation of large volumes of PRP, used mainly in orthopedic surgery. Although the design (e.g., inclusion of leukocytes) can vary, these machines eliminated biases due to operators and reduced individual variations. However, they also increased patients’ physical burden and economic load. Thus, these devices are not widely used in clinical settings in many countries, except the United States.

In contrast, several PRP preparation kits (e.g., Ycellbio) (Ycellbio Medical Co., Ltd., Seoul, Korea) (Figure 1b) [54] have been developed in various countries. In addition to reducing individual variations, the most significant advantage of these kits is their applicability at small sample volumes. Thus, these kits can be used in regenerative dentistry without burdening patients. However, leukocytes also become concentrated in the resulting PRP preparations due to their design concept. As such, this type of preparation kit should be used, depending on the clinical case.

### 3.3. Brand Competition and Consumer Report-Like Study in the Subsequent Generation

Since Choukroun’s study on PRF, liquid and clotted PRF have also been modified to produce derivatives. Among the derivatives, A-PRF and CGF are the most popular. Very recently, another brand was developed, BIO-PRF, as introduced by Miron [55] (Section 3.3.1). The idea of horizontal centrifugation to produce the PRF was introduced by Lourenço et al. [56]. Additionally, the PRF prepared using conventional horizontal centrifuges and glass tubes, without modification on the surface could not be considered as an innovative procedure and distinguished. It is another branded preparation protocol that requires their specific devices and has been competing with other companies to occupy the market. These competitions in themselves could be seen as the result of economic activity. However, from a biomedical point of view, it is unreasonable that, despite using specific protocols, PRF preparations prepared by devices supplied by third parties cannot be accepted as genuine, brand-specific PRF preparations. For example, the angulation of the rotor significantly influences the quality (e.g., fibrin architecture and retention ability) of growth factors and, thus, of the resulting PRF preparations (Section 3.3.1). Some “minor” differences have been observed in the architecture and platelet distribution of PRF derivatives prepared while using different types of centrifuges [30,57,58,59]. However, it has not yet been demonstrated clearly whether these differences significantly influence clinical outcomes. Thus, excessive, non-scientific brand competitions not only hamper advances in PRF research and therapy, but also distort the therapeutic significance of PCs. However, despite non-scientific, brand-based critiques of our research activity, we have obtained significant centrifuges (Figure 2).

In contrast, PRP has not faced competition based on economic interests, despite many expensive automated preparation machines competing to occupy the market. Instead, to date, less-scientific, consumer report-like comparative studies have often been published to nominate machines with the best performance. Unfortunately, such data have rarely been endorsed by clinical outcomes in PRP.

Therefore, clinicians must learn more about the classification and terminology of PRP/PRF derivatives [60] and carefully opt for any derivatives on the basis of specific biomedical data. To date, many clinicians have opted for derivatives without carefully considering the literature or conducting fair comparisons.

#### 3.3.1. Specificity of Centrifuge Types

In terms of PRF derivatives, because L-PRF and A-PRF are often discussed, we will cover BIO-PRF here. BIO-PRF was first reported in 2018 [56] and, despite not being a popular option, horizontal centrifugation facilitates conversion while using fixed-angle centrifuges for PRF preparation. Often used in PRP preparation, horizontal spinning allows for blood samples to be resolved in density gradients and promotes the effective separation of individual blood cells, while fixed angle rotors are useful for a variety of applications, from pelleting blood cells to the isopycnic separation of macromolecules [56,61]. Because we retrieved the upper fraction of PRF, horizontal centrifuges are theoretically more suitable. Furthermore, horizontal centrifuges reduce the probability of cell-cell and cell-inner wall collision, thereby preventing accelerated cell adhesion and potential injury. In fact, horizontal centrifuges at higher speeds are able to recover platelets in PRF matrices at higher levels [55].

In clinical practice, the PRF matrix is separated from the red blood cell fraction (i.e., red thrombus). In this step, clinicians use scissors or spatulae and more or less invade the region of the PRF matrix. In this case, when simply comparing the cross-sectional areas, PRF prepared using horizontal centrifuges are the smallest (Table 2). For example, the cross-sectional area of the PRF matrix that was prepared using the Intra-Spin centrifuge was 1.83-fold larger than that prepared using horizontal centrifuges (Table 2). Thus, the loss of platelets can be theoretically minimized using a horizontal centrifuge. 

In PRF matrices, because platelets are not distributed by density gradients, the platelets are not the most accumulated component around the interface. Thus, regardless of the operators’ skill, many platelets are hardly lost during mechanical separation of RBC fraction. Because of the reduced probability of blood cell collision, the collision-induced activation of platelets was suppressed, and platelet entrapment was increased [55]. However, the superiority of the horizontal centrifuge is not yet demonstrated clearly in PRF preparation as in PRP preparation.

#### 3.3.2. Specificity of Blood Collection Tube Types

PRF quality is significantly influenced by the type of tube used. However, this has not been extensively studied, since the tubes used for PRF preparation are mainly those that are produced for blood testing. Recently, brand manufacturers/vendors have been providing genuine tubes for blood collection. In many cases, the materials and elements used in the fabrication of the tubes are not fully disclosed. This is important, since, for example, silicone coating on glass surfaces is known to markedly reduce the adhesion of blood cells and the adsorption of plasma proteins. Additionally, surface modification can significantly prolong coagulation time (Kawase et al., unpublished observations), a phenomenon that is thought to depend on the composition of silicone derivatives and the level of contaminants.

Plastic tubes coated with silica microparticles are convenient for blood coagulation in blood testing. Kawase and his group demonstrated that this PRF matrix can be distinguished from that prepared by glass tubes, and that silica microparticles incorporated into the PRF matrix (Figure 3) pose a health hazard [62,63]. Against these clear scientific indications, the corresponding vendor still claims that they are safe and continues selling silica-coated plastic tubes for PRF preparation without disclosing potential health risks. Thus, we recommend that clinicians carefully examine blood-collection tubes prior to setting up their preparation system and the vendor discloses the safety data if any.

In contrast to the thoughtful, positive modifications made to improve PRF quality, it is not yet known how these unconscious, negative modifications may influence clinical outcomes. Therefore, from a safety point of view, clinicians should be aware of these differences and take the different factors into careful consideration.

### 3.4. Proposed Mechanisms or Modes of Action of PC in PC Therapy

From the initial phase of the PRP study, various attempts have been made to clarify the mechanisms or modes of PRP action. PRP therapy was introduced into the field of regenerative therapy initially assuming that platelets and their bioactive factors are highly concentrated. Subsequent in vitro and in vivo studies provided evidence for this assumption and supported its clinical use [18,26,27,64,65]. However, the regenerative action of PRP is not solely induced by growth factors, but by various factors that are contained in PRP [4,66] preparations. Insoluble fibrin, which is converted from soluble fibrinogen upon activation, functions not only as a scaffolding material, but also as a carrier of growth factors to potentiate growth factor action in an additive or synergistic manner by delaying growth factor degradation [67,68]. In contrast, anticoagulants, such as sodium citrate and EDTA, and coagulation factors, such as calcium chloride and thrombin, can positively or negatively influence the proliferation and differentiation of cells that are involved in tissue regeneration [69,70,71,72,73]. This classic concept is illustrated in Figure 4a: PRP initially and primarily acts through growth factors/cytokines on circulating hematopoietic stem cells (even in low quantity) and endothelial progenitor cells to form new blood vessels to facilitate the supply of cells, oxygen, and nutrients to regenerating sites. In addition to this direct action, PRP directly and indirectly acts on cells that are involved in tissue regeneration in collaboration with other contents, such as fibrin [65]. Unknown factors, which could be leukocytes, endogenous proteases, and anticoagulants or coagulation factors, may hinder these regenerative processes. In the case of young, healthy patients who have sufficient activity of spontaneous regeneration, it is thought that these individual “players” act cooperatively according to this scenario.

In contrast, in the case of elderly, less healthy patients whose regenerative activity is suppressed, the appropriate surgical operation or medication is needed to regain the spontaneous regenerative activity and induce conditions sensitive to subsequent PRP therapy. We speculate that several unknown factors or mechanisms may suppress the spontaneous regenerative activity behind these phenomena (Figure 4b). If the surgical operation and medication are recognized as “major players”, PRP therapy could be interpreted as adjuvant therapy. In contrast, assuming that PRP is the main therapeutic factor, surgery and medication could be denoted as “conditioning therapies”, and PRP therapy as “replacement therapy”. It should be noted that “unknown factor X” does not necessarily specify a certain compound, but rather broadly includes factors that range from compounds to pathological conditions. 

### 3.5. RNA Delivery System

As described above, the PC therapy has mainly been developed taking growth factors into account, and various biomolecules contained in PCs are thought to positively or negatively modulate the action of a majority of the identified growth factors. In addition to this classic concept, RNA delivery theory was recently proposed and investigated as a new concept in understanding PC action. The platelet-derived extracellular vesicle (EV), which is composed of exosomes and micro-vesicles, was first described by Wolf in 1967 [74]. However, the role of this component was thought to be “platelet dust” and it was not investigated for a long period of time until now [75]. EVs contain messenger RNA, microRNA, long-coding RNA, and circular RNA [76] besides concentrated growth factors, such as bFGF, VEGF, PDGF-BB, and TGF-β1 [77]. Unlike the receptor-ligand interaction that was observed in growth factors, these RNAs are internalized into recipient cells to modify their behaviors. In tissue regeneration, it is thought that EVs can confer proangiogenic, proliferative, antiapoptotic and anti-inflammatory properties on the recipient cells [75,78]. Thus, there is no doubt that the regenerative property of PCs partially depends on these small components. In fact, this new concept has been employed to develop novel therapeutic strategies targeting angiogenesis for tissue regeneration [79] by not having to depend on the use of intact stem cells or platelets.

### 3.6. Controlled Release of Growth Factors Retained in PRF Matrices

In recent years, the controlled release of growth factors in PRF matrices has been extensively investigated and compared between PRF derivatives in in vitro experimental systems [30,80,81]. From a technological point of view, the obtained data from these studies have improved the characterization and our understanding of the use of individual PRF derivatives as biomaterials. However, PRF matrices implanted in tissues enriched with blood vessels, such as subcutaneous connective tissue, are degraded rapidly (within 1–2 weeks) [82,83,84]. In addition, because the experimental systems are not reconstituted with plasmin or other endogenous proteases, studies have not simulated in vivo conditions. Thus, clinicians should not overestimate the corresponding in vitro data. The ability of prolonged retention and delayed release of growth factors is hardly indicative of biomedical significance in vivo at the site of implantation.

With regards to this limitation of PRF, Kawase et al. [85] established the so-called heat-compression technique as a way to reduce the biodegradation of PRF in 2015. They found that the heating could modify the native PRF membrane to be used as a barrier for guided bone regeneration procedures. Similar to this concept, Mourão et al. [86] conducted an in vitro study, in which they heated the blood serum and a portion of plasma low in platelets, subsequently incorporating PRP or liquid PRF for the inclusion of cells; consequently, the possibility of releasing growth factors and other cytokines. At the time, the product was called Alb-CGF, corresponding to the albumin produced by the heating process, with the incorporation of the concentrate of growth factors [86,87]. Recently, the same group carried out new studies in vitro in order to assess the biocompatibility of this method [88]. In this study, they changed the name to Alb-PRF to distinguish this method from liquid PRF [87,88]. They also performed an in vivo study, in which they observed biodegradation in mice, and observed the Alb-PRF membrane after 21 days in subcutaneous tissue, indicating its slow degradation and the potential use of this blood by-product as a barrier [89]. Although further studies will be needed to assess the behavior of these autologous barriers, the development of such a highly functionalized PRF matrix will facilitate the discussion of the controlled release of growth factors from PRFs in vivo and its clinical significance.

### 3.7. Lack of Biological Consideration of Platelets

Besides these trends and major topics, various minor topics have been individually or independently investigated. However, somehow, platelet biology has rarely been investigated in reference to their involvement in the preparation, and application of PCs. In the history of PC study and therapy, platelets have been considered almost exclusively as carriers of growth factors, but rarely as living cells that act as a multifunctional minimum biological unit. Indeed, it has often been demonstrated that frozen or freeze-dried PRP, in which no living platelets or other blood cells are observed, retains its ability to facilitate cell proliferation and wound healing in in vitro and in vivo experimental systems [18,26,27,64]. Therefore, despite having significant knowledge about growth factors that are derived from platelets, several clinicians and researchers in this field, including us, do (did) not sufficiently understand platelet biology.

For example, it is sometimes mentioned in hands-on seminars or technical notes that PRP and platelets are “activated” by coagulation factors to form a fibrin clot in the final step of the preparation process prior to clinical use. However, because such a phenomenon is not anticipated in our body under pathophysiological conditions, there have been few published articles reporting Ca^2+^-induced platelet activation. Thus, we interpreted such a process as a conceptual, but not evidence-based, expression, and examined this possibility in a previous study [90]. Kawase and his group successfully found that exogenously added Ca^2+^ directly activated platelets in order to facilitate adhesion to a titanium surface and release microparticles and growth factors. To fill gaps in their knowledge, these researchers have further accumulated basic knowledge related to platelet biology in a series of studies [72,91,92,93,94,95,96]. However, many platelet functions that are closely related to PC therapy are still poorly understood and have not yet been clarified.

Platelet biology has comprehensively been summarized and updated in previous review articles [97,98,99]. Therefore, if unfamiliar with this topic, readers are expected to refer to these publications. Despite their limited relevance to PC therapy, several main and supporting functions of platelets, such as aggregation [100,101,102,103], adhesion [103,104,105], activation [102,106,107], growth-factor delivery and roles [108,109], anti-bacteria [110,111], pain-relief [112,113], coagulation [106,107,114,115,116], and interaction with leukocytes [117,118], should be investigated in more detail in order to understand how living platelets act during preparation and therapeutic processes. Significant advances in understanding platelet biology will enable us to optimize PC preparation further and improve the predictability and quality of PC therapy.

## 4. Overlooked Clinical Studies

During the initial phases of PC study, many clinical studies were conducted at various levels. In the past decade, upon the request of the regulatory authorities, randomized controlled trials (RCT) have been conducted in cooperation with several core hospitals using larger sample sizes. However, to our knowledge, rigorous evidence has not yet been provided to satisfy the regulatory authorities, mainly due to a lack of standardization or sophisticated design [119,120]. Judging from the principle of the classic RCTs [121], it is easy to see why evidence has not yet been obtained. However, judging from the principle of the recently proposed concept, namely pragmatic RCT [122,123], we can see that negative evidence has been obtained. In Table 3, classic and pragmatic RCT are compared to past and current RCTs for PRP/PRF therapy.

In classic RCT, project leaders should carefully design the study plan to (1) collect homogenous participants, (2) reduce biases by eliminating variations, and (3) detect medically important differences by setting the original primary endpoint. In contrast, to evaluate the quality of RCTs, reviewers should pay attention to (1) the size of type I error α (usually 0.05), which represents the probability of a conclusion that treatments are different when, in fact, they are really equivalent, (2) the power (usually 0.80 or 0.90) or β, which represents the probability of a conclusion that the treatments are not different when in fact they are different (type II error), and (3) the sample size necessary to achieve this desired precision [120]. The typical size of a phase III RCT is 100 to 1000 patients. In a multicenter RCT for FGF-2 that was conducted ten years ago in Japan under the advice of the regulatory authorities [124], 253 patients with periodontitis were originally enrolled and allocated into four groups. The statistical power was 0.90 and the two-sided type I error rate was 0.025 (for comparison of each FGF-2 group with the “vehicle alone” group), according to the sample size calculation. Thus, the study design and data were theoretically sound and of high quality.

In terms of PRP/PRF preparation, for better homogeneity, PRP/PRF samples should be prepared by well-trained operators using the same devices according to the standardized protocols (Figure 5). At present, unfortunately, there are no practical or theoretical solutions available to reduce inevitable variations in individual samples and assure the quality of the resulting PRP/PRF preparations quickly prior to clinical use. Thus, this is the biggest fundamental bias, which can be categorized as a “performance bias” (Table 4), in PRP/PRF clinical trials, and it is not observed in the usual clinical trials of new drugs. However, it is theoretically possible to minimize this bias by standardizing the preparation protocols and developing new technologies in the near future. The systemic and local conditions of patients are not necessarily examined from the viewpoint of regenerative medicine. 

However, the more serious issue is a poor RCT design. To date, although many RCTs have been conducted to support the clinical use of PCs, the majority have been empirically designed and they have rarely been examined for their quality. Although several meta-analyses have indicated poor quality [119,126], many meta-analyses have been performed on the data obtained from such poor RCTs. Moreover, it is suspected that a significant percentage of these meta-analyses, regardless of whether they are published or not, have been based on the expectation that PCs are promising biomaterials in regenerative therapy. This phenomenon is known as cognitive bias [127], and we should be careful in referring to such publications.

Undoubtedly, this situation is theoretically and scientifically inappropriate. However, these data may be of some worth. Pragmatic RCTs do not require a homogenous sample population, and the population is more similar to that used in clinical practice. As for the quality of the PRP/PRF preparations, potential biases may be minimized by standardization. However, in this concept, the outcomes should be accepted as they are. In addition, because many clinical studies of PCs, including RCTs, use small sample sizes or the wrong durations, we should also consider potential publication biases in the literature [128,129,130]. Pannuci and Wilkins explained other potential biases in research [125] (Table 4). Thus, without careful consideration of the degree of the biases, using data obtained from RCTs for meta-analysis is risky, and the meta-analysis results may lead to misunderstandings. Overall, the previous findings indicate that PCs may not be effective, at least in terms of their use in bone regeneration. However, this will have to be analyzed further in future studies.

Regardless of the type of RCT, many steps still need to be standardized with regards to the preparation and clinical application of PCs. Although it may be practically challenging to conduct testing of all PC preparations prior to their use [7], we should reconsider the requirements of PC therapy in the clinical setting to plan further laboratory research.

## 5. Prioritized Research Investigations

In general, RCTs for PC therapy should be conducted according to the CONSORT guideline, as are RCTs for other biomaterials [131], and biases that are not specifically related to PCs should be excluded or minimized. Further, studies should be carefully designed. 

We believe that the principle of the RCT has been widely misunderstood for some time. It is commonly found that increasing the sample size solves most problems and provides clear evidence, which may be related to the recent emphasis on big data. However, this option is limited in the case of PC therapy. Unsuccessful clinical trials should be systematically analyzed, evaluating appropriate pathological conditions of recipients (e.g., availability of stem cells, the ability of angiogenesis, levels of inflammation, or activity of balanced bone metabolism).

Additionally, standardizing the preparation protocols and developing convenient and straightforward point-of care testing are expected to improve PC quality. Furthermore, case-specific, suitable partner cells for use in PC therapy and the high functionalization and modification of PC preparations are also expected to improve PC quality (Section 3.5). Table 5 outlines some of the steps that will need to be taken in future studies, which we believe should be cooperatively carried out in order to allow for the development of better PC therapy.

## 6. Conclusions

Mainly, from a cost-effective point of view, PC therapy has been excessively used without careful quality assurance of PC preparations or careful examination of recipient conditions. However, PCs are not the exceptional, wonder-drugs of regenerative medicine. PC therapy is thought to exert regenerative efficacy under conditions that maintain spontaneous regenerative activity, or it is improved by preceding or parallel surgical operation or medication. Thus, in most cases, the latter combination treatments are required to reproducibly obtain significant clinical outcomes in PC therapy. However, ironically, such a combination treatment makes it difficult to conduct high-quality RCTs and obtain clear evidence in order to terminate the endless controversy. Thus, what we, as clinicians and researchers, can or should do for future PC therapy is to standardize the quality, protocols, and diagnosis. We believe that basic bioengineering studies must enable realizing the standardization of this important therapeutic strategy.

## Figures and Tables

**Figure 1 bioengineering-07-00082-f001:**
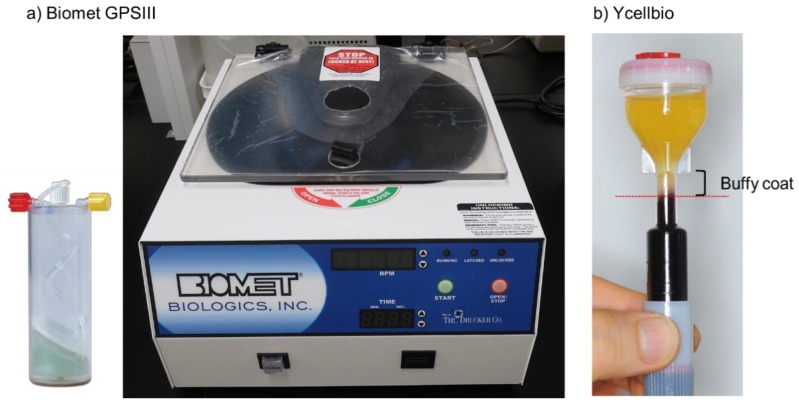
(**a**) Biomet GPSIII platelet concentration system and (**b**) Ycellbio System [54].

**Figure 2 bioengineering-07-00082-f002:**
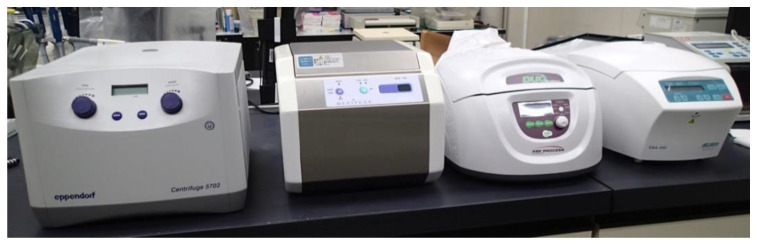
Major centrifuges used for PRF preparation. From left to right: Eppendorf #5702 (for BIO-PRF), Medifuge (for CGF), Duo Quattro (for A-PRF), and Hettich EBA200 (original model of Intra-Spin) (for L-PRF).

**Figure 3 bioengineering-07-00082-f003:**
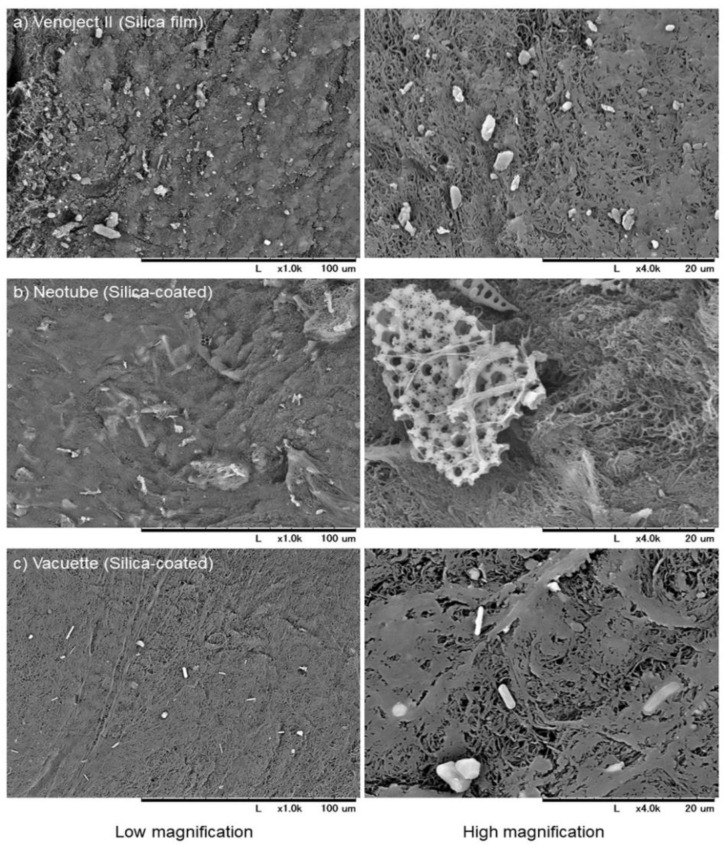
SEM observations of silica microparticles embedded in and attached to individual A-PRF-like matrices prepared using plastic tubes containing silica microparticles, (**a**) Venoject II, (**b**) Neotube, and (**c**) Vacuette. Scale bars are presented at the bottom of each photo [63].

**Figure 4 bioengineering-07-00082-f004:**
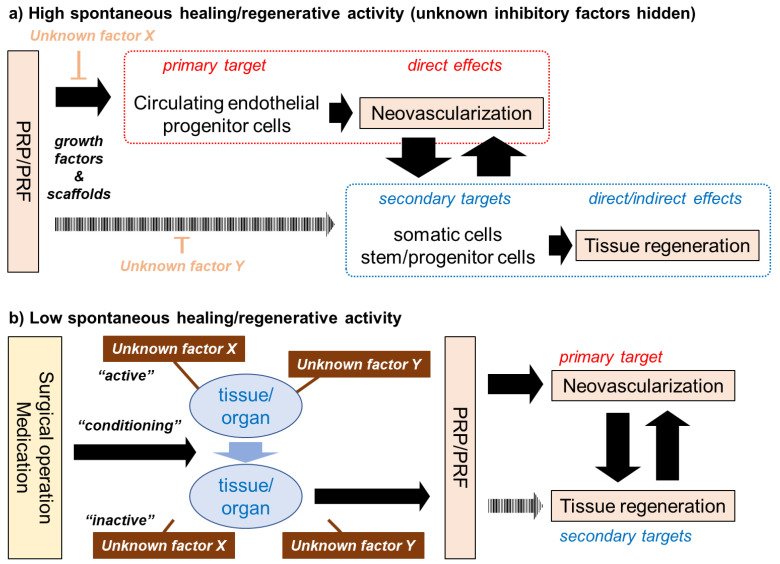
Mechanisms of PC (platelet-rich plasma (PRP)/PRF) action on tissue regeneration. (**a**) The generally accepted concept, which is thought to follow high levels of spontaneous regenerative activity. (**b**) Our newly proposed concept, might may follow low levels of spontaneous regenerative activity.

**Figure 5 bioengineering-07-00082-f005:**
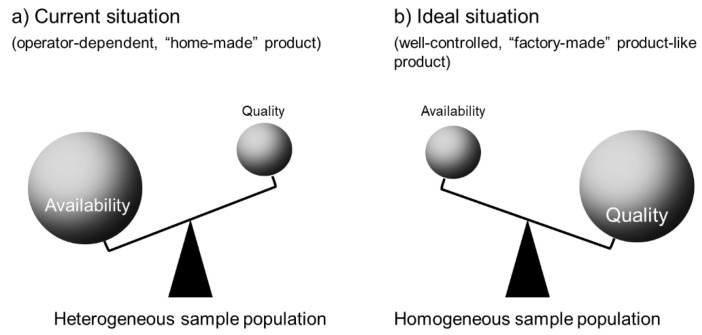
Homogeneity of PRP/PRF samples. (**a**) Current situation: PRP/PRF samples are “home-made” according to the individual operators’ standards. Almost all preparations are used for regenerative therapy but are not discarded at lower qualities. Thus, the quality can vary between individual samples and the population is considered “heterogenous”. (**b**) Ideal situation: similar to well-controlled, “factory-made” products, PRP/PRF samples are prepared using standardized protocols and their quality is individually inspected against standard criteria. Thus, many samples are clinically excluded and discarded. However, this quality control makes the population highly “homogeneous”.

**Table 1 bioengineering-07-00082-t001:** History of fundamental platelet concentrate (PC) study.

Generation	Major Contents
1st	Validation of concentrated platelets and growth factors Development of automated PRP preparation devices
2nd	Modification of preparation protocols and developments of novel PRP derivatives, such as platelet-rich fibrin (PRF)
3rd	Comparisons of PRP derivatives: ability to retain and release growth factors, mechanical strength, biodegradability, etc.
4th	Exploration of coupling partner cells of PCs

**Table 2 bioengineering-07-00082-t002:** Angulation of rotors used in centrifuges and cross-sectional areas of the interface between PRF and RBC fractions.

	Intra-Spin	Duo Quattro	Horizontal Type
Angulation (°)	33.0	41.3	90.0
Cross-sectional area (mm^2^)	183.6π	151.5π	100.0π
Shape of cross-section	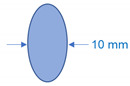	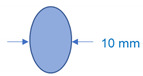	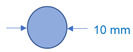
Images of angulation and cross-section	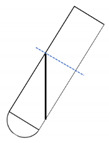	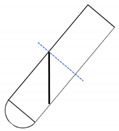	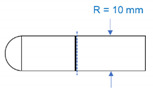

**Table 3 bioengineering-07-00082-t003:** Comparison of classic randomized controlled trials (RCT), pragmatic RCT, and past and current RCT for PRP/PRF therapy.

Criteria	Classic RCT	Past and Current PRP/PRFs RCTs	Pragmatic RCT
**Required by**	Regulatory authorities	–	Clinicians
**Patients/cases**	Homogeneous	Heterogeneous	Heterogeneous
**Combinational treatments**	Controlled	Non- or less-controlled	Less-controlled
**PC preparation protocol**	Highly standardized	Less-standardized	Less-standardized
**PC quality**	Protocol-dependent	Not assured	Not assured
**Sample size**	Large (typically 100–1000; power-test at least 80%)	Small ~ middle	Small ~ large
**Research center**	Preferential multicentric	1 ~ several	1 ~ several
**Primary end points**	Narrow (single)	Narrow ~ middle (multi)	Middle (multi)?
**Clinical outcomes**	–	Controversial (negative ≥ positive)	–

**Table 4 bioengineering-07-00082-t004:** Various types of potential biases in RCTs and tips to avoid such biases [125]. This table has been reproduced with permission from Wolters Kluwer.

Type of Bias	How to Avoid
**I.** **Pretrial bias**	
Flawed study design	Clearly define the risks and outcomes, preferably with an objective or validated method. Standardize and blind data collection.
Selection bias	Select patients using rigorous criteria to avoid confounding results. Patients should be sourced from the same general population. Well designed, prospective studies help to avoid selection bias since the outcome is unknown at the time of enrollment.
channeling bias	Assign patients to study cohorts using rigorous criteria.
**II.** **Bias during trial**	
Interview bias	Standardize interviewer’s interaction with patient. Blind interviewer to exposure status.
Chronology bias	Prospective studies can eliminate chronology bias. Avoid using historic controls (confounding by secular trends).
Recall bias	Use objective data sources whenever possible. When using subjective data sources, corroborate with medical records. Conduct prospective studies because the outcome is unknown at the time of patient enrollment.
Transfer bias	Carefully design plan for lost-to-follow-up patients prior to the study.
Exposure Misclassification	Clearly define exposure prior to study. Avoid using proxies of exposure.
Outcome misclassification	Use objective diagnostic studies or validated measures as the primary outcomes.
Performance bias	Consider cluster stratification to minimize variability in surgical technique.
**III.** **Bias after trial**	
Citation bias (publication bias)	Register trial with an accepted clinical trials registry. Check registries for similar unpublished or in-progress trials prior to publication.
Confounding	Known confounders can be controlled with study design (case control design or randomization) or during data analysis (regression). Unknown confounders can only be controlled with randomization.

**Table 5 bioengineering-07-00082-t005:** Major goals for determining the efficacy of PCs.

Stage	Major Requirements
Basic and preclinical	Standardization of PC preparation protocols
	Standardization of PC quality criteria
	Standardization of PC shipping criteria
	Exploration of partner cells suitable for combinational PC treatment
	Development of highly functionalized PC derivatives
Clinical	Standardization of criteria of indications
	Standardization of PC therapeutic protocols
	Definition of responder and non-responder

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
