# Peer review of "The Platelet Concentrates Therapy: From the Biased Past to the Anticipated Future"

_bioengineering, 2020, doi:10.3390/bioengineering7030082_

Round 1

Reviewer 1 Report

The manuscript entitled ‘Study of platelet concentrates’ is an interesting review/opinion article, which presents a unique outlook on the clinical use of platelet concentrates from scientific, methodical, commercial and social-science points of view. It is very well-written, convincing and is easy to follow. I have only minor comments that may improve the impact of this review:

  1. The study title is too generic and may have more impact if the authors emphasised the most critical issue they want this review to be remembered for. For example, a similar review was entitled: Use of platelet lysate for bone regeneration-are we ready for clinical translation?
  2. The abstract does not cover the main unique views presented in the review, for example history of PRP (four generations), concepts of adjuvant therapy versus primary therapy, the importance of centrifuge rotors and collection tubes, and the rigorous design of clinical trials. As it stands, the abstract is rather generic and similar to previous reviews published in the PC field.
  3. Their proposed model for PC mechanisms of action for recipients with low innate regenerative capacity is interesting, but it doesn’t encompass the fact that both PCs and autologous stem cells are likely to be suboptimal in these situations. Do the authors propose using allogeneic PCs (with or without allogeneic stem cells) in these patients?
  4. Given increasing interest in extracellular vesicles (EVs), the authors are encouraged to comment on any emerging data on EVs in PCs.
  5. It would be nice to have a separate table where the authors clearly present their views and supporting literature on ‘suitable partner cells for use in PC therapy’.
  6. What would be, in the authors’ opinion, most suitable animal models to study the effects of ‘unknown factors’ such as endogenous proteases, on PC’s performance in vivo, i.e. those that can’t be modelled in standard in vitro experiments? Given we need to reduce the use of animals of research, would any 3D models or tissue organoids be suitable for this purpose?

Author Response

Comments and Suggestions for Authors

1) The manuscript entitled ‘Study of platelet concentrates’ is an interesting review/opinion article, which presents a unique outlook on the clinical use of platelet concentrates from scientific, methodical, commercial and social-science points of view. It is very well-written, convincing and is easy to follow. I have only minor comments that may improve the impact of this review:

The study title is too generic and may have more impact if the authors emphasised the most critical issue they want this review to be remembered for. For example, a similar review was entitled: Use of platelet lysate for bone regeneration-are we ready for clinical translation?

Response: Thank you for your advice. We have modified the title as recommended by you.

2) The abstract does not cover the main unique views presented in the review, for example history of PRP (four generations), concepts of adjuvant therapy versus primary therapy, the importance of centrifuge rotors and collection tubes, and the rigorous design of clinical trials. As it stands, the abstract is rather generic and similar to previous reviews published in the PC field.

Response: As per your recommendation, we have included these topics and have rewritten the abstract.

3) Their proposed model for PC mechanisms of action for recipients with low innate regenerative capacity is interesting, but it doesn’t encompass the fact that both PCs and autologous stem cells are likely to be suboptimal in these situations. Do the authors propose using allogeneic PCs (with or without allogeneic stem cells) in these patients?

Response: Thank you for this interesting question. As you are aware, due to possible transmission risks of pathogens, it is not recommended to use allogeneic PCs in clinical settings. However, a new project is now underway to reconstitute PCs from platelet-derived allogeneic iPS cells in combination with medical-grade fibrinogen, albumin, and other major plasma proteins. Besides, various types of allogeneic stem cells are commercially available for medical use or under clinical trials for approval. Thus, we think it is theoretically possible (approval will be taken) to combine allogeneic PCs and allogeneic stem cells for regenerative therapy in the near future. In this case, we can choose candidates having the potential to create PC therapy of the highest quality.

4) Given increasing interest in extracellular vesicles (EVs), the authors are encouraged to comment on any emerging data on EVs in PCs.

Response: We are grateful for this comment. We have added a new subsection (3.5.) regarding the involvement of EVs in PC action in regenerative therapy.

5) It would be nice to have a separate table where the authors clearly present their views and supporting literature on ‘suitable partner cells for use in PC therapy’.

Response: This is a good point. However, we do not have enough information to make a separate table in the article. To the best of our knowledge, mesenchymal stem cells derived from bone marrow and adipose tissue are major partner cells. Our groups have been studying autologous tissue-engineered cells sheets and using these biomaterials in combination with PRP [Yamamiya et al. J Periodontol 79,811-818,2008; Nagata et al. Bone 50(5),1123-1129,2012]. Thus, we think that periosteal cells, of which the majority are osteoblast precursors, are suitable to be used as a combination with PRP. However, such adult cell types are “not standard” at present. When many more other cell types are proposed for combination use, we hope to release a systematic review.

6) What would be, in the authors’ opinion, most suitable animal models to study the effects of ‘unknown factors’ such as endogenous proteases, on PC’s performance in vivo, i.e. those that can’t be modelled in standard in vitro experiments? Given we need to reduce the use of animals of research, would any 3D models or tissue organoids be suitable for this purpose?

Response: This is a very important point in further developing the PC study. We have been thinking for quite some time of ways by which such studies could be incorporated. We speculate that some types of glycoproteins, which are involved in cell senescence, or something similar may reduce cellular responses to growth factors related to PCs. Thus, we hope to initiate in vitro studies to clarify this mechanism.

Reviewer 2 Report

It was a pleasure to review this interesting manuscript authored by Kawase et al. with the title “Study of platelet concentrates”. The topic is of critical importance and suits perfectly for Bioengineering. However, the manuscript requires major improvements and corrections and may not be accepted unless the issues are addressed. The reviewer has concerns about the submitted version of the manuscript as indicated below.

Major issues without any specific order

(1) To scientifically evaluate and report the effects of PCs in an evidenced-based manner, the authors should adopt a systematic approach to analyze the performed studies on PC and perform a meta-analysis, so that this review can be clinically relevant and the goal of the manuscript can be reached.
(2) The “ideal situation” of the authors was defined as “factory-made” product-like product” (P11L366). This requires detailed explanation and discussion; or should be removed and the manuscript must be revised where necessary. First, the preparations are autologous, patients are not the same, and their platelets/their therapeutic activity are not the same. Second, the centers follow a protocol, so it is not the operator’s protocol.
(3) The conclusion is not acceptable and should be re-written. The language should be more scientific. Several phrases are vague and should be eliminated or explained. The conclusion should be synthesized in an evidence-based manner. Also, the last sentence is awkward, it is not clear what should be improved in the authors’ understanding of PC exactly, what exactly is unknown, and what original studies do the authors propose to address it.
(4) PL38-39: “In this review article, we look back on the recent history of PC research, reconfirm the ultimate goals of this field…” It should be made clear how the authors look back, and how old is exactly the “recent history”, last 5 years? If so, all published papers were screened and analyzed? The methods regarding this review manuscript are not clear.
(5) Platelet biology is not included in the manuscript.
(6) The voice of the manuscript must be intensively revised throughout the manuscript; including but not limited to the use of vague phrases, unclear or subjective terms. (For authors to consider: Also, the Reviewer personally prefers to avoid the word “we” in scientific manuscripts.)
(7) P1L36-37: This sentence does not reflect the reality, thus, should be removed.
(8) The manuscript lacks some critical information and should be added. What is the proposal of the authors to address the standardization of PC preparation protocols, quality/shipping criteria?
(9) Table 4 is exactly taken from the study of Pannucci et al. (Table 1 in the original study) Thus, instead of re-printing the same table in this manuscript, the authors should direct the reader to the original study.
(10) The manuscript should also discuss the benefit of standardization while it is known that the diseases and patients are very different from each other.
(11) P12L389: What does “carefully” mean?
(12) P5L198: This does not seem correct.
(13) P1L11: “ideal” is vague. Ideal for the treatment of which conditions of which patients?
(14) P1L20: It is not clear what “these conditions” mean.
(15) Some relevant papers, reviews, systematic reviews were not cited.
(16) P15L407: It is not clear what do authors mean with “successful”.

Minor issues without any specific order
(17) The title is too general and should be more specific.
(18) Can the authors add a clear sentence in the introduction about which therapies/applications there are / might be considered under PC therapy?
(19) Mechanisms or modes of action of PC therapies are very important, therefore, the reviewer prefers it to be a separate section (not under “History”).
(20) Figs. 1 and 2 can be combined into one figure.
(21) Reference 29 lacks issue/volume/page information.
(22) P12L402: “targets” can be confusing, maybe “goals” or “objectives” or “aims” can be a better choice of word.

Author Response

Comments and Suggestions for Authors

It was a pleasure to review this interesting manuscript authored by Kawase et al. with the title “Study of platelet concentrates”. The topic is of critical importance and suits perfectly for Bioengineering. However, the manuscript requires major improvements and corrections and may not be accepted unless the issues are addressed. The reviewer has concerns about the submitted version of the manuscript as indicated below.

Major issues without any specific order

1) To scientifically evaluate and report the effects of PCs in an evidenced-based manner, the authors should adopt a systematic approach to analyze the performed studies on PC and perform a meta-analysis, so that this review can be clinically relevant and the goal of the manuscript can be reached.

Response: Thank you for your relevant comment. A systematic approach could have been taken, if the PC study and therapy had been researched and developed as thoroughly as other major drugs.

However, we believe that the biggest problem behind its considerably slow progress is the careless meta-analysis that dealt with data obtained from poorly designed clinical trials. In our opinion, sound and reliable clinical trials of PC therapy cannot be conducted without an assurance of PC quality and recipient conditions. Thus, the accumulated clinical data of PC therapy is undoubtedly helpful for improving the current PC study and therapy; however, it cannot be evaluated as evidence to convince the regulatory authorities about the clinical effectiveness of PC preparations.

2) The “ideal situation” of the authors was defined as “factory-made” product-like product” (P11L366). This requires detailed explanation and discussion; or should be removed and the manuscript must be revised where necessary. First, the preparations are autologous, patients are not the same, and their platelets/their therapeutic activity are not the same. Second, the centers follow a protocol, so it is not the operator’s protocol.

Response: In this figure legend (Fig. 5), we intend to mention the necessity and importance of quality control and assurance system besides standard operating procedures (SOP), i.e., standardized protocols in PC preparation. As you may be aware, in European Union, CE marking is provided to “factory-made” products that meet the requirements of standardization for the whole manufacturing process, including quality assurance. CE marking does not ensure individual quality; however, once poor quality is found, this approval is canceled.

As for pharmaceutical products, in the United States, FDA approves the manufacturing process, quality, safety, and effectiveness of individual medicines. We indicate such established systems as “ideal situations”. Whatever the protocol is, either an operators’ or a standardized one, it is one of requirements for approval and to secure minimum levels of product quality; however, it does not assure the total quality of a particular product. We think that we have explained this situation in details in this portion. However, if the readers have no or poor knowledge of manufacturing and approval systems, it may be difficult to convince them without an introduction to these systems.

3) The conclusion is not acceptable and should be re-written. The language should be more scientific. Several phrases are vague and should be eliminated or explained. The conclusion should be synthesized in an evidence-based manner. Also, the last sentence is awkward, it is not clear what should be improved in the authors’ understanding of PC exactly, what exactly is unknown, and what original studies do the authors propose to address it.

Response: According to your comment, we have extensively revised the Conclusion section.

4) PL38-39: “In this review article, we look back on the recent history of PC research, reconfirm the ultimate goals of this field…” It should be made clear how the authors look back, and how old is exactly the “recent history”, last 5 years? If so, all published papers were screened and analyzed? The methods regarding this review manuscript are not clear.

Response: This is not a systematic review rather a narrative review or expert opinion that analyzes trends subjectively. According to us, frankly speaking, few poor systematic review articles have distorted the history of PC studies and misled many clinicians. This kind of negative history has rarely been published because “evidence” is not available in the literature (i.e., a kind of publication bias). However, when we perform hands-on seminars or some other ventures that provides opportunities to closely talk with clinicians, we have always found that many clinicians misunderstand or overestimate PC therapy. We guess that you may have similar experiences.

Moreover, we have analyzed some representative basic studies receiving high citations or review counts to extract trends and significant contribution to the current PC therapy. Unfortunately, the current PC clinical studies seem to not be in tune with basic studies. In other words, many basic studies have been performed according to the authors’ curiosity but are not closely related to the needs of clinical settings. We deplore this situation although you may require us to present the evidence of such situations.

If PCs had been studied and progressed like “factory-made” drugs, basic and preclinical studies could have led clinical studies to have proper directions and the approval of the regulatory authorities in earlier phases (or not). This is the “evidence” for the poor situation of PC studies.

5) Platelet biology is not included in the manuscript.

Response: As you have indicated, platelet biology is important to correctly understand what PCs are. However, platelets have usually been considered as carriers of growth factors and their functions of controlled release are studied. In addition, if we include this topic, the pages will exceed 30 pages and 20,000 words. Readers may not correctly understand what we want to claim or it may be cumbersome to read, which is against our expectations. We have continuously investigated platelet biology and published several articles; therefore, we have not included platelet biology in this article.

6) The voice of the manuscript must be intensively revised throughout the manuscript; including but not limited to the use of vague phrases, unclear or subjective terms. (For authors to consider: Also, the Reviewer personally prefers to avoid the word “we” in scientific manuscripts.)

Response: We understand your point. However, as we have mentioned elsewhere in these responses to comments, this is not a systematic review based predominantly on the analyses of previously published articles. This is a narrative review to subjectively show our personal opinions. Thus, to avoid readers misunderstanding our objective, we thought that we place an emphasis on “we” instead of using the passive voice. We hope you will understand our standpoints.

7) P1L36-37: This sentence does not reflect the reality, thus, should be removed.

Response: This underlines our pessimistic perspective and depicts the worst scenario. We have not discussed the current situation or the reality but have highlighted possible ways we can go about to avoid such a worse scenario. We hope you will understand the purpose of this specific statement for the subsequent development.

8) The manuscript lacks some critical information and should be added. What is the proposal of the authors to address the standardization of PC preparation protocols, quality/shipping criteria?

Response: We must emphasize that this is a narrative review and expert opinion. Some parts are discussed based on the literature, but some are merely our opinions or proposals. We think that some articles are cited for standardization of PC preparation protocols. In contrast, to the best of our knowledge, there is no standardization of quality/shipping criteria. In Japan, PC therapy is included into the new regulatory framework for regenerative therapy and the quality of PC preparations should be assured prior to shipping. However, unfortunately, the regulatory authorities or the concerned medical societies have not yet provided convincing criteria or guidelines to clinicians. Thus, it is necessary to have such criteria based on relevant evidence.

9) Table 4 is exactly taken from the study of Pannucci et al. (Table 1 in the original study) Thus, instead of re-printing the same table in this manuscript, the authors should direct the reader to the original study.

Response: We also thought the option that you have suggested. However, we have re-printed the same table because we know that many clinicians (readers) usually do not access the original article. The additional purpose of this table is to get readers to intuitively understand the way by which many biases influence randomized controlled trials and how carefully RCT should be designed to avoid such biases.

10) The manuscript should also discuss the benefit of standardization while it is known that the diseases and patients are very different from each other.

Response: As described in Table 4 and related paragraphs, standardization is important to minimize potential biases. However, as we have repeatedly mentioned, it is not advisable to assure the quality of individual PC preparations because individual PC preparations and recipients (patients) are different. Thus, we have continuously asserted the necessity of point-of-care testing for assurance of individual PC quality as described in a previous article [Kawase et al., Int J Growth Factors Stem Cells Dent 2,13-17,2019]. We do not highly evaluate the benefits of standardization as other researchers do. For us, the standardization is something that is better than nothing. Therefore, we have not made a space for this discussion.

11) P12L389: What does “carefully” mean?

Response: As far as we have checked, the past RCTs, including our own RCTs, of PC therapy lacked “carefulness” in the biases related to PC preparations. In the discussion, the differences have always been attributed to individual differences in PC preparations. In this article, we want to claim that there are many more biases that influence the data and quality of RCT. Thus, we have suggested that RCTs should be designed carefully.

12) P5L198: This does not seem correct.

Response: We think that commercial competition is not so bad in PRP as observed in PRF because such automated machines for PRP preparations require large volumes of blood samples and are used in limited fields of therapy. However, we did not mention that there is no competition among venders who market PRP preparation machines. Venders have been competing to increase the market share. In fact, we have peer-reviewed and read consumer report-like articles of PRP preparation machines. Thus, we understand that it may reflect existence of the competition.

13) P1L11: “ideal” is vague. Ideal for the treatment of which conditions of which patients?

Response: We have reworded this with “more predictable.”

14) P1L20: It is not clear what “these conditions” mean.

Response: The “conditions” mean that evidence is required for approval of the regulatory authorities. We have reworded it.

15) Some relevant papers, reviews, systematic reviews were not cited.

Response: We have cited more than 100 references that are closely relevant to the outline of this article. To the best of our knowledge, there are only a few available references that discuss mainly biases and tips to overcome the designing of clinical trials of PC therapy. This article also has a purpose to enlighten clinicians and researchers who are not familiar with the “dark side” of PC studies, but are only concerned about the “sunny side” of PC study, e.g., individual differences of PC preparations. Thus, we think that our article is highly original. Unfortunately, we have not come across relevant review articles that may strongly impact this article.

16) P15L407: It is not clear what do authors mean with “successful”.

Response: In the Conclusion section, the first “successful” means “PC therapy providing significant clinical outcomes. The second “successful” means “high-quality”. We have reworded these words.

Minor issues without any specific order

17) The title is too general and should be more specific.

Response: Thank you for this comment. This article is not an overview of the past PC studies but our suggestions for future PC therapy. Thus, we have modified the title as follows: The platelet concentrates therapy: from the biased past to the anticipated future.

18) Can the authors add a clear sentence in the introduction about which therapies/applications there are / might be considered under PC therapy?

Response: We have added the medical fields and treatments in which PCs are used.

19) Mechanisms or modes of action of PC therapies are very important, therefore, the reviewer prefers it to be a separate section (not under “History”).

Response: We have modified the title of this subsection as “Proposed mechanisms or modes of action of PC in PC therapy” because we have discussed not only the mechanism of PC action, but also the response of recipient tissues and cells in this subsection.

Basically, we agree with your opinion that the mechanisms or modes of PC action is very important. In fact, several efforts have been made to clarify the mechanisms. Consequently, many factors have been identified that are involved positively or negatively in PC action. However, it is thought that there are some more unidentified factors in PCs that modify PC action. Assuming that most of these factors crosstalk with one another to a more or less extent, it is impossible to draw individual dose-response curves and clearly identify the roles of individual factors. It should be noted that according to the advice of Reviewer 1, we added the subsection of RNA delivery system, another mechanism of PC action that has recently received much attention.

Thus, we again claim that it is important to clarify the mechanisms of PC action. However, at the same time, we are afraid that this effort may not necessarily facilitate the approval of PC therapy by the regulatory authorities. We should try to learn much more about recipient conditions. Moreover, we should concentrate more on clinical studies to obtain evidence to convince the regulatory authorities. Of course, basic bioengineering studies are expected to be useful to standardize the procedures and develop methods to ensure the quality of individual preparations and their clinical use.

According to the reviewer’s advice, we have modified the title of this article. What we want to appeal to readers is that we should prioritize what to do to establish a more predictable PC therapy in the future. In this context, we have not placed a priority on the project to clarify the mechanism of PC action. Thus, it is difficult to treat “mechanism” in a separate section.

20) Figs. 1 and 2 can be combined into one figure.

Response: We have combined these figures into one figure.

21) Reference 29 lacks issue/volume/page information.

Response: This article was published online but not in the printed version. Thus, volume and page numbers are not yet provided. Instead, we added the DOI number.

22) P12L402: “targets” can be confusing, maybe “goals” or “objectives” or “aims” can be a better choice of word.

Response: Thank you for this advice. We chose to write it as “goals.”

Round 2

Reviewer 2 Report

The reviewer has seen the responses of the authors to the reviewer’s comments. Below are reviewer comments that the authors are encouraged to consider. The impact of the manuscript will increase if the authors include 1 (one) paragraph on the platelet biology, and cite suitable references on the platelet biology (Comment 5 in the first reviewer report). To improve the manuscript, if possible, the authors may consider to revise/adapt and include some of their responses to the first reviewer report to the suitable places in the manuscript (for example, Last sections of the responses to the 1st and 2nd comments in the first reviewer report). As the authors know, it should be noted that the figures or tables that were previously published without an open-access license, need an obtained permission from the copyright holder to make them open access in a new publication, or else they cannot be reused. The situation of all the figures and tables in the manuscript should be confirmed.

Author Response

Reviewer 2

The reviewer has seen the responses of the authors to the reviewer’s comments. Below are reviewer comments that the authors are encouraged to consider. The impact of the manuscript will increase if the authors include 1 (one) paragraph on the platelet biology, and cite suitable references on the platelet biology (Comment 5 in the first reviewer report). To improve the manuscript, if possible, the authors may consider to revise/adapt and include some of their responses to the first reviewer report to the suitable places in the manuscript (for example, Last sections of the responses to the 1st and 2nd comments in the first reviewer report).

As the authors know, it should be noted that the figures or tables that were previously published without an open-access license, need an obtained permission from the copyright holder to make them open access in a new publication, or else they cannot be reused. The situation of all the figures and tables in the manuscript should be confirmed.

Response: Thank you for your advice. It was difficult to find an appropriate place to insert “platelet biology” without sacrificing the context and disturbing readers’ understanding. However, we inserted this topic along with a significant number of references at the end of section #3. We spent three paragraphs for this subsection but limited the expansion as much as we could. We hope that you will be satisfied with this revision.

As for the responses to the comments made by Reviewer #1, we reworded and inserted our thoughts of allogeneic PRP and partner cells into the section #3 (in the second generation).

As for the reuse of figures and tables, both Fig. 1b and Fig. 3 are derived from open-access journals. Regarding Table 4, we are now requesting the publisher for permission. Please see the attached PDF file.

Round 3

Reviewer 2 Report

My overall recommendation is a conditional acceptance until the copyrighted material can be re-used in an open access manner.